# Recent Advances in Combination of Immunotherapy and Chemoradiotherapy for Locally Advanced Esophageal Squamous Cell Carcinoma

**DOI:** 10.3390/cancers14205168

**Published:** 2022-10-21

**Authors:** Ruixi Wang, Shiliang Liu, Baoqing Chen, Mian Xi

**Affiliations:** 1State Key Laboratory of Oncology in South China, Collaborative Innovation Centre for Cancer Medicine, Guangdong Esophageal Cancer Institute, Guangzhou 510060, China; 2Department of Radiation Oncology, Sun Yat-sen University Cancer Center, No. 651 Dongfeng East Road, Guangzhou 510060, China

**Keywords:** esophageal squamous cell carcinoma, immune checkpoint inhibitors, immunotherapy, chemoradiotherapy, locally advanced

## Abstract

**Simple Summary:**

Neoadjuvant chemoradiotherapy (CRT), followed by surgery or definitive CRT, is the standard treatment for locally advanced esophageal squamous cell carcinoma (ESCC); however, the clinical outcomes remain unsatisfactory. Immunotherapy combined with CRT is currently being investigated as a novel treatment option for locally advanced ESCC. In this review, we discuss the theoretical background and status of immunotherapy for locally advanced ESCC and potential biomarkers for predicting tumor response and prognosis.

**Abstract:**

Esophageal cancer has a high mortality rate and a poor prognosis, with more than one-third of patients receiving a diagnosis of locally advanced cancer. Esophageal squamous cell carcinoma (ESCC) is the dominant histological subtype of esophageal cancer in Asia and Eastern Europe. Although neoadjuvant or definitive chemoradiotherapy (CRT) has been the standard treatment for locally advanced ESCC, patient outcomes remain unsatisfactory, with recurrence rates as high as 30–50%. The combination of immune checkpoint inhibitors (ICIs) and CRT has emerged as a novel strategy to treat esophageal cancer, and it may have a synergistic action and provide greater efficacy. In the phase III CheckMate-577 trial, one year of adjuvant nivolumab after neoadjuvant CRT improved disease-free survival in patients with residual disease on pathology. Moreover, several phase I and II studies have shown that ICIs combined with concurrent CRT may increase the rate of pathologic complete response for resectable ESCC, but they lack long-term follow-up results. In unresectable cases, the combination of camrelizumab and definitive CRT showed promising results against ESCC in a phase Ib trial. Phase III randomized trials are currently ongoing to investigate the survival benefits of ICIs combined with neoadjuvant or definitive CRT, and they will clarify the role of immunotherapy in locally advanced ESCC. Additionally, valid biomarkers to predict tumor response and survival outcomes need to be further explored.

## 1. Introduction

Esophageal cancer is the seventh most common cancer and the sixth leading cause of cancer-related deaths worldwide. Globally, 604,100 new cases of esophageal cancer were diagnosed, and 544,076 deaths were reported in 2020 [1]. There are two different histological types of esophageal cancer: esophageal squamous cell carcinoma (ESCC), which is more common in East Asian and Middle Eastern countries, and esophageal adenocarcinoma (EAC), which has a higher incidence in Western countries [2].

Over 50% of ESCC cases are at a locally advanced stage when diagnosed. Neoadjuvant or definitive chemoradiotherapy (CRT) has been the standard treatment for locally advanced ESCC, but the clinical outcomes of patients remain unsatisfactory, with five-year overall survival (OS) rates ranging from 25% to 47% [2]. The recurrence rates after CRT can be as high as 30–50%. However, there is no established strategy for the adjuvant treatment of patients with esophageal cancer at a high risk of recurrence after standard therapy. 

In recent years, immune checkpoint inhibitors (ICIs) have led to significant improvements in clinical outcomes across many malignancies, including esophageal cancer [3,4,5,6,7]. The administration of programmed death receptor 1 (PD-1) inhibitor monotherapy has demonstrated superior anti-tumor activity compared with physician’s choice chemotherapy in second-line therapy for advanced esophageal cancer with manageable safety, according to the results from ESCORT, KEYNOTE-181, and ATTRACTION-3 studies [5]. Notable studies, such as ESCORT-1st, KEYNOTE-590, and CheckMate648, have advanced the exploration of first-line immunotherapy for advanced cases and have demonstrated significant efficacy. Based on the results from these randomized phase III trials, ICIs in combination with chemotherapy have been recommended as the first-line treatment instead of chemotherapy alone for advanced esophageal cancer [3,4,5,6,7]. In locally advanced cases, the combination of ICIs and CRT has emerged as a novel strategy with a possible synergistic action and greater efficacy. Here, we review the clinical application of ICIs in patients with locally advanced ESCC and discuss potential biomarkers for predicting tumor response and survival outcomes.

## 2. Mechanism of PD-1/PD-L1 Monoclonal Antibody in Combination with CRT

Programmed death ligand 1 (PD-L1), also known as B7-H1 or CD274, belongs to the immunoglobulin B7 family. Under physiological conditions, PD-L1/PD-1 is a synergistic inhibitory signaling pathway in which PD-L1 binds to the receptor PD-1 on activated T cells, maintaining peripheral tolerance and immune homeostasis in T cells, thereby preventing T cell over-activation to avoid autoimmune disease [8]. The PD-1/PD-L1 interaction, which is frequently observed in human malignancies, suppresses the activation of T cells, allowing tumors to successfully elude anti-tumor immunity. Thus, PD-1 expression on T cells combined with PD-L1 expression in tumor and tumor-infiltrating immune cells is crucial in the tumor microenvironment and has exceptional clinical importance [9]. PD-1 or PD-L1 monoclonal antibodies have shown clinical anti-tumor activity by activating T cell immune function to block the synergistic PD-L1/PD-1 inhibitory signaling pathway [10].

In anti-tumor therapy, combining chemotherapy with ICIs has a synergistic effect [11]. Chemotherapeutic agents activate a specific adaptive immune response against tumors by inducing immunogenic cell death [12]. In addition, chemotherapeutic agents can act on cells in the tumor microenvironment to activate anti-tumor immunity.

By enhancing antigen presentation (including human leukocyte antigens), CD8 expression, and DNA damage, radiotherapy causes considerable immunomodulation that results in type I interferon responses, pro-inflammatory effects, and T-cell-mediated immunogenic killing [13]. In combination with immunotherapy, radiotherapy kills tumor cells, thereby promoting the release of tumor-specific antigens and enhancing the acquired immune response. Radiotherapy generates a systemic anti-tumor immune response by activating antigen-presenting dendritic cells, which act synergistically with the sustained immune activation of immunotherapy to induce T cell homing. Moreover, increasing evidence indicates that combining radiotherapy and immunotherapy can strengthen abscopal effects [14].

## 3. ICIs for Locally Advanced Resectable ESCC

### 3.1. ICIs Combined with Neoadjuvant CRT

Several phase I and II studies have investigated the safety and efficacy of PD-1 inhibitors combined with neoadjuvant CRT in ESCC. In addition, several ongoing trials have explored the benefits of this combination (Table 1).

In a single-arm phase II clinical trial (NCT02844075) reported at the 2019 American Society of Clinical Oncology (ASCO) Annual Meeting, 28 patients with locally advanced ESCC received a combination of neoadjuvant pembrolizumab and CRT (paclitaxel + cisplatin), followed by surgery and postoperative pembrolizumab maintenance up until two years, disease progression, or intolerable toxicity [15]. With a median follow-up of 11.7 months, the pathological complete response (pCR) rate was 46.1%, and the one-year OS rate was 89.3%.

Similarly, in a phase I study (PALACE-1), preoperative pembrolizumab combined with neoadjuvant CRT resulted in an acceptable safety profile and anti-tumor efficacy in 20 patients with locally advanced ESCC [16]. Grade 3 or higher treatment-related adverse events occurred in 65% of the enrolled patients, with lymphocytopenia accounting for most of these occurrences (92%). The pCR rate was 55.6% in 18 patients who underwent surgery, which was higher than that of historical results from traditional neoadjuvant CRT [17,18]. A subsequent ongoing multicenter phase II clinical study (PALACE-2) will further confirm the efficacy of this treatment [19].

A phase II single-arm trial evaluating neoadjuvant CRT in combination with perioperative toripalimab for ESCC was presented at the ASCO Annual Meeting in 2022 (NCT04437212) [20]. In 13 patients who underwent the procedure, the overall pCR rate was 54%, and 7 patients (54%) experienced grade 3–4 treatment-related adverse events.

In addition, SCALE-1 was a single-center, single-arm, exploratory phase Ib study that enrolled 23 patients with locally advanced ESCC. A short course of neoadjuvant radiotherapy combined with toripalimab was used, with a total radiation dose of 30 Gy in 12 fractions and two cycles of paclitaxel/carboplatin chemotherapy administered concurrently. After surgery, 11 patients exhibited pCR (47.8%), suggesting adequate safety and efficacy of short-course CRT combined with immunotherapy. 

Collectively, the available data suggest that the combination of immunotherapy and neoadjuvant CRT in ESCC has the potential to achieve a higher rate of pCR and manageable safety, but it lacks long-term follow-up results. More evidence from phase III trials is required to confirm these findings.

### 3.2. ICIs Combined with Neoadjuvant Chemotherapy

Although neoadjuvant CRT is the standard treatment for resectable locally advanced ESCC, it is also associated with higher postoperative complications and mortality than surgery alone. Neoadjuvant chemotherapy and nivolumab showed encouraging results, including a major pathologic response (MPR) rate of 83% in resectable lung cancer in the NADIM study [21]. Therefore, several researchers have investigated the efficacy of neoadjuvant treatments combining PD-1 inhibitors and chemotherapy in ESCC, excluding radiotherapy (Table 2). 

A prospective study of neoadjuvant camrelizumab in combination with paclitaxel and carboplatin (ChiCTR2100051903) reported pCR and R0 resection rates of 31.3% and 93.8%, respectively [22]. After a one-year follow-up, progression-free survival (PFS) and OS were 83% and 90.9%, respectively, with manageable adverse events.

Neoadjuvant chemotherapy has poor outcomes in patients with multisite lymph node metastases, and the NICE study is the first to focus on neoadjuvant therapy in this cohort globally. This prospective, single-arm, single-center, exploratory phase II trial assessed the effectiveness and safety of camrelizumab combined with carboplatin and albumin paclitaxel (ChiCTR1900026240) [23]. This study showed a pCR rate of 39.2% and complete remission of the primary tumor in five patients with residual lesions in the lymph nodes (ypT0N+). As the long-term efficacy of the NICE regimen is lacking, a phase III randomized controlled trial is underway.

Toripalimab plus neoadjuvant chemotherapy (paclitaxel + carboplatin) showed anti-tumor effectiveness in locally advanced ESCC in a phase II trial (NCT04177797) [24]. The MPR and pCR rates were 43.8% and 18.8%, respectively, in the 16 patients who underwent surgery.

FRONTiER is a phase I clinical study that aims to evaluate the safety of nivolumab in combination with neoadjuvant chemotherapy regimens (5-fluorouracil + paclitaxel or 5-fluorouracil + docetaxel) [25]. The ASCO Gastrointestinal Cancer Symposiums Annual Meeting in 2021 and 2022 reported the preliminary results of the FRONTiER study, which showed that two patients (33.3%) in cohort A achieved pCR, and the pCR rates in cohorts C and D were 16.7% and 50.0%, respectively [26,27]. Only one patient in cohort D had dose-limiting toxicity (grade 3 dyspnea and rash).

In previous studies, patients usually received concurrent chemotherapy and anti-PD-1 monoclonal antibody treatment; however, chemotherapy drugs can kill anti-PD-1 antibody-activated T cells, hindering the effects of the ICI [28]. A recent phase II study showed that administering toripalimab 48 h after administering chemotherapy (paclitaxel + cisplatin) resulted in a higher pCR rate than concurrent administration (NCT03985670) [29].

In summary, neoadjuvant PD-1 inhibitors combined with chemotherapy may be effective against resectable ESCC. However, the superiority of this modality to neoadjuvant CRT remains unclear and warrants further investigation.

### 3.3. Adjuvant Immunotherapy after Neoadjuvant CRT and Surgery

Recurrence risk remains significant for patients with esophageal cancer who have residual disease after neoadjuvant CRT followed by surgery. Until recently, high-level evidence-based recommendations for adjuvant treatment were lacking. Currently, a novel adjuvant nivolumab maintenance therapy after surgery is recommended in these cases, based on the results of the CheckMate 577 trial.

CheckMate-577 is a randomized, double-blind, multicenter phase III study that enrolled 1085 patients who had esophageal or gastroesophageal junction cancer (ESCC 29%) without pCR after neoadjuvant CRT [30]. Patients were randomized in a 2:1 ratio to receive adjuvant nivolumab or placebo. The primary endpoint of disease-free survival (DFS) was reached. According to the findings, the median DFS for the 532 patients who received nivolumab was 22.4 months, which was significantly longer than the 11.0 months for the 262 patients in the placebo group. Additionally, compared with 15 (6%) of 260 patients in the placebo group, 71 (13%) patients in the nivolumab group experienced treatment-related grade 3 or 4 adverse events. The findings of this trial have now been endorsed by treatment guidelines, which indicate that adjuvant treatment with nivolumab significantly prolongs DFS compared with placebo and has an adequate overall safety profile. However, a clinical study conducted by Park et al. in 2022 (European Society for Medical Oncology—ESMO) contrasted with Checkmate577 (NCT02520453) [31]. This placebo-controlled, randomized, double-blind, phase II clinical study included 86 ESCC patients undergoing neoadjuvant chemoradiotherapy, staged T3-4N0M0 or T1-4N1-3M0 according to the American Joint Committee on Cancer (AJCC) 7th edition, and it evaluated the efficacy of adjuvant durvalumab compared to placebo. There was no statistically significant difference between the two groups in terms of DFS or OS after a median follow-up of 38.7 months. 

CheckMate-577 was a multicenter phase III study with a large sample size and excluded patients who achieved pCR postoperatively. Conversely, the study conducted by Park et al. was a relatively small single-center trial that enrolled both postoperative pCR and non-pCR patients. With a higher recurrence risk after trimodality therapy, patients with non-pCR were more likely to benefit from adjuvant therapy. Therefore, the negative results from the Korean phase II trial are to be expected.

Additionally, a multicenter, randomized, controlled phase III clinical study (AIRES) is currently ongoing to assess the efficacy and safety of adjuvant chemotherapy combined with tislelizumab versus tislelizumab alone for patients with ESCC at high risk of postoperative recurrence (ypN+). 

## 4. Immunotherapy in Locally Advanced Unresectable ESCC

### 4.1. ICIs Combined with Definitive Radiotherapy

For patients with locally advanced esophageal cancer who cannot tolerate concurrent CRT, definitive radiotherapy alone is the standard choice. A phase Ib trial explored the safety and feasibility of radiotherapy combined with camrelizumab as a first-line treatment in 20 patients with locally advanced ESCC who were ineligible for or refused concurrent CRT [32]. Patients underwent 54–60 Gy (1.8–2.0 Gy/dose) radiation and were also administered camrelizumab (200 mg, Q2W) for 32 weeks. The median OS and PFS were 16.7 and 11.7 months, respectively. This study highlighted the controlled toxicity and anti-tumor efficacy of radiotherapy combined with camrelizumab.

Similarly, the American Society for Radiation Oncology (ASTRO) 2018 Annual Meeting reported another phase II trial analyzing the efficacy and safety of camrelizumab in combination with radiotherapy in locally advanced ESCC [33]. The results showed that 6 patients (42.9%) experienced grade 1–2 treatment-related adverse events, with 1 (7.1%) and 13 (92.9%) exhibiting complete response and partial response, respectively. These two non-randomized trials support the safety and feasibility of combining radiotherapy and immunotherapy, although the data were limited to a small number of patients. Another clinical study investigating definitive radiotherapy in combination with immunotherapy is ongoing (NCT03200691).

### 4.2. ICIs Combined with Definitive CRT

Definitive CRT is currently the standard of care for locally advanced unresectable esophageal cancer. It is unclear whether combined immunotherapy can improve therapeutic effectiveness. To date, the published clinical data in this setting are limited to two studies, but several phase III randomized multicenter trials are ongoing and will help to identify the role of immunotherapy against esophageal cancer (Table 3).

A recent phase II clinical trial evaluated the efficacy of durvalumab and tremelimumab in conjunction with definitive CRT (5-fluorouracil + cisplatin) in patients with unresectable locally advanced ESCC [34]. The results showed that the PFS and OS at 24 months were 57.5% and 75%, respectively. A subgroup analysis showed that patients with PD-L1-positive tumors (*n* = 28) had significantly longer PFS and OS than those with PD-L1-negative tumors. Notably, only one patient had a grade 4 adverse event. The preliminary findings indicate that durvalumab and tremelimumab have promising efficacy and safety profiles when compared with a historical control group. Additionally, PD-L1 expression demonstrated a high predictive value in the study population.

In another phase Ib trial, camrelizumab in combination with apatinib and concurrent CRT (docetaxel + cisplatin) in 20 patients with locally advanced ESCC showed that radiation esophagitis (20%) and esophageal fistula (10%) were the most common treatment-related grade 3 adverse events [35]. OS and PFS ranged from 8.2–28.5 months and 4.0–28.5 months, respectively. The 12- and 24-month OS rates were 85.0% and 69.6%, respectively, while the PFS rates were 80.0% and 65.0%, respectively. 

Taken together, these two small prospective studies suggest that the combination of immunotherapy and definitive CRT could improve survival and toxicity compared with traditional CRT against ESCC. High-quality evidence from randomized trials is expected to optimize patient outcomes.

## 5. Predictors of Efficacy for Tumor Immunotherapy

In most studies on advanced esophageal cancer, PD-L1 is a useful but imperfect biomarker for response to immunotherapy. These studies include KEYNOTE-181, ESCORT-1st, KEYNOTE-590, and CheckMate648 [3,4,5,6]. However, the predictive value of PD-L1 expression in patients with locally advanced esophageal cancer is controversial. According to the subgroup analysis of the NCT02520453 study, the three-year OS was longer with durvalumab treatment than with placebo in the PD-L1 combined positive score (CPS) ≥ 1% group (94% versus 64%), but it was shorter in the PD-L1-negative group (42% vs. 55%) [31]. Nevertheless, in the CheckMate-577 study, nivolumab treatment showed similar clinical benefits regardless of tumor cell PD-L1 expression, indicating that this biomarker was not predictive of clinical benefit in this trial [30]. In addition, PALACE-1 researchers found that TCF-1 + CD8+ T cell infiltration was considerably higher in pCR tumors; however, PD-L1 expression was not related to pathological remission rates.

Ma et al. reported the spatial distribution of dendritic cells and macrophages in a phase Ib trial (NCT03671265) of definitive CRT combined with camrelizumab for unresectable ESCC, and they assessed the association between spatial distribution, outcome, and tumor mutational load (TMB) [36]. Multiplex immunofluorescence assays were performed to identify tumor CD11c+ dendritic cells, CD68+ macrophages, and PD-L1 expression levels at baseline and during treatment, and whole-exome sequencing was performed to assess TMB. The findings revealed considerable spatial distribution of PD-L1− or PD-L1+ dendritic cells and macrophages in ESCC. Additionally, there was a close correlation between high TMB and shorter distances between tumor cells, DCs, and macrophages. This trial verified the geographical pattern of adenomatous polyposis coli (APC) and its ability to predict outcomes for radiation and immunotherapy. 

Another study (ChiCTR2000028900) revealed that the number of PD-L1+ CD163+ tumor-infiltrating lymphocytes was significantly lower in a pCR group than in a non-pCR group after treatment with camrelizumab combined with neoadjuvant CRT (*p* = 0.017) [37]. M2-like macrophages are specifically identified by the biomarker CD163, and research has shown that M2-like macrophages with elevated PD-L1 expression can promote ICIs.

Previous studies have shown that high levels of tumor-infiltrating lymphocytes (TILs) in the tumor microenvironment are associated with a survival advantage in non-small-cell lung cancer, melanoma, cervical cancer, and head and neck squamous cell carcinoma [38,39,40,41]. TILs are a heterogeneous group of lymphocytes that infiltrate tumors and participate in anti-tumor responses [42]. In a recent phase II trial of toripalimab for the treatment of Chinese patients with locally advanced ESCC, CD8+ TIL density was considerably greater in responders following neoadjuvant immunotherapy (NCT04177797). In contrast, CD8+ TIL density did not significantly increase in non-responders. Collectively, the predictive value of TILs in ESCC requires further investigation.

## 6. Conclusions and Future Directions 

For many years, the cornerstone of multimodal therapy for locally advanced ESCC has been CRT and/or surgery; however, a considerable percentage of patients exhibit a worsening clinical profile. At present, immunotherapy is also an important part of the treatment and management of ESCC, either in advanced stages or in locally advanced stages. Promising outcomes have been noted in published studies. Clinical evidence indicates that CRT combined with immunotherapy may be a novel and encouraging therapeutic option for patients with locally advanced ESCC. However, due to the lack of long-term follow-up results, the role of this new combination needs to be further established.

In the future, the optimal combination modalities to treat locally advanced ESCC should be investigated. Clinical research should adapt and optimize CRT regimens, as well as individualize stratified therapy. ICIs in combination modalities will likely benefit more patients, as the anti-tumor immune properties of PD-1/PD-L1 inhibitors in esophageal cancer become clearer. Moreover, the reduction in toxicity associated with combination therapy is essential to maximize survival. Further research is necessary to identify optimal biomarkers that can detect the patients most likely to benefit from immunotherapy.

## Figures and Tables

**Table 1 cancers-14-05168-t001:** Immunotherapy combined with neoadjuvant chemoradiotherapy for resectable esophageal squamous cell carcinoma: ongoing trials.

Registration	Phase	Study Cohort	Control Cohort	Primary Endpoints	Secondary Endpoints
NCT04435197 (PALACE-2)	II	Pembrolizumab + CRT (Paclitaxel + Carboplatin)		pCR	DFS, OS
NCT04437212	II	Toripalimab + CRT (Paclitaxel + Cisplatin)		MPR	DFS, OS, AEs
NCT03044613	Ib	Nivolumab + CRT (Paclitaxel + Carboplatin)	Nivolumab + Relatlimab + CRT (Paclitaxel + Carboplatin)	AEs	pCR, OS, RFS
NCT03064490 (PROCEED)	II	Pembrolizumab + CRT (Paclitaxel + Carboplatin)		pCR	AEs
NCT04006041	II	Toripalimab + CRT (Paclitaxel + Cisplatin)		pCR	OS, DFS, AEs, R0 resection rate
NCT04177875	II	Toripalimab + CRT (Docetaxel + Nab-paclitaxel + Cisplatin)		MPR, ORR	DFS, OS, AEs
NCT04229459	II	Nivolumab + Cetuximab + CRT (Cisplatin + 5-FU)		pCR, PFS, AEs	OS
NCT04568200	II	Durvalumab + CRT (Paclitaxel + Carboplatin)	Placebo + CRT (Paclitaxel + Carboplatin)	Tumor response, pathological response	R0 resection rate, AEs, PFS, OS
NCT04644250	II	Toripalimab + CRT (Paclitaxel liposome + Carboplatin)		pCR	AEs, OS, DFS, MPR, ORR, R0 resection rate
NCT04776590	II	Tislelizumab + CRT (Nab-paclitaxel + Carboplatin)		pCR	DFS, OS
NCT04888403	II	Toripalimab + CRT (Nab-paclitaxel + Nedaplatin)		pCR	MPR, DFS, R0 resection rate
NCT04929392	II	Pembrolizumab + Lenvatinib + CRT (Nab-paclitaxel + Carboplatin)		pCR, cCR	AEs, DFS, OS
NCT04974047	II	Tislelizumab + CT (Paclitaxel + Cisplatin)	Tislelizumab + CRT (Paclitaxel + Cisplatin/5-FU + Cisplatin)	pCR	MPR, R0 resection rate, DFS, EFS, ORR, AEs
NCT03165994	II	Sotigalimab + CRT (Paclitaxel + Carboplatin)		pCR	R0 resection rate, pathologic stage, radiographic/metabolic response, AEs
NCT03490292	I/II	Avelumab +CRT (Paclitaxel + Carboplatin)		DLT, pCR	AEs, DFS, R0 resection rate
NCT03857763	II	Apatinib + CRT (Paclitaxel + Cisplatin)		pCR	R0 resection rate, DFS, OS, AEs
NCT04426955	III	Camrelizumab + CRT (Paclitaxel + Cisplatin)	Placebo + CRT (Paclitaxel + Cisplatin)	PFS	OS, ORR, DoR, AEs

CRT, chemoradiotherapy; pCR, pathological complete response; DFS, disease-free survival; OS, overall survival; MPR, major pathologic response; AEs, adverse events; RFS, recurrence-free survival; ORR, objective response rate; PFS, progression-free survival; cCR, clinical complete response; EFS, event-free survival; CT, chemotherapy; DLT, dose-limiting toxicity; DoR, duration of response.

**Table 2 cancers-14-05168-t002:** Immunotherapy combined with neoadjuvant chemotherapy for resectable esophageal squamous cell carcinoma: ongoing trials.

Registration	Phase	Study Cohort	Control Cohort	Primary Endpoints	Secondary Endpoints
NCT04280822 (HCHTOG1909)	III	Toripalimab + CT (Paclitaxel + Cisplatin)	Placebo + CT (Paclitaxel + Cisplatin)	EFS	pCR, DFS, OS, ORR, R0 resection rate, MPR, AEs, QoL
NCT04506138	II	Camrelizumab + CT (Nab-paclitaxel + Carboplatin)		pCR, MPR	OS, EFS
NCT04177797	II	Toripalimab + CT (Paclitaxel + Carboplatin)		pCR	DCR, ORR, AEs
NCT03914443 (FRONTiER)	I	Nivolumab + CT (5-FU + Cisplatin) (Cohort A/B)	Nivolumab + CT (5-FU + Cisplatin + Docetaxel) (Cohort C/D)	DLT	pCR, AEs, PFS, OS
NCT04389177 (KEYSTONE-001)	II	Pembrolizumab + CT (Paclitaxel + Carboplatin)		MPR	ORR, DFS, OS, R0 resection rate
NCT03437200 (CRUCIAL)	II	Nivolumab + CT (FOLFOX)	Nivolumab + Ipilimumab + CT (FOLFOX)	PFS	FFS, OS
NCT04460066	Ib/II	ZKB001 + CT (Nab-paclitaxel + Cisplatin)	Placebo + CT (Nab-paclitaxel + Cisplatin)	MPR	R0 resection rate, pCR, DFS, EFS, OS, AEs
NCT04767295	II	Camrelizumab + CT (Nab-paclitaxel + Carboplatin)		pCR	OS, PFS, DCR, ORR
NCT04804696	II	Toripalimab + CT (Paclitaxel + Cisplatin)		pCR	ORR, R0 resection rate, MPR, DFS, OS, AEs
NCT04848753	III	Toripalimab + CT (Paclitaxel + Cisplatin)	Placebo + CT (Paclitaxel + Cisplatin)	EFS	pCR, EFS, OS
NCT03917966	II	Camrelizumab + CT (Docetaxel + Nedaplatin)	Camrelizumab + Apatinib	MPR	pCR, OS, DFS, LNR, R0 resection rate
NCT03946969	II	Sintilimab + CT (Nab-paclitaxel + Cisplatin + S-1)		AEs	MPR, R0 resection rate, RFS, OS

CT, chemotherapy; EFS, event-free survival; pCR, pathological complete response; DFS, disease-free survival; OS, overall survival; ORR, objective response rate; MPR, major pathologic response; AEs, adverse events; QoL, quality of life; DCR, disease control rate; DLT, dose-limiting toxicity; PFS, progression-free survival; FFS, failure-free survival; LNR, lymph node ratio; RFS, recurrence-free survival.

**Table 3 cancers-14-05168-t003:** Immunotherapy combined with radical chemoradiotherapy for unresectable esophageal squamous cell carcinoma: phase III ongoing trials.

Registration	Phase	Population Studied	Study Cohort	Control Cohort	Primary Endpoints	Secondary Endpoints
NCT03957590 (RATIONALE-311)	III	ESCC	Tislelizumab + CRT (Paclitaxel + Cisplatin)	Placebo + CRT (Paclitaxel + Cisplatin)	PFS	ORR, DoR, OS, AEs
NCT04210115 (KEYNOTE-975)	III	ESCC/EAC/GEJC	Pembrolizumab + CRT (Cisplatin + 5-FU or FOLFOX)	Placebo + CRT (Cisplatin + 5-FU or FOLFOX)	OS, EFS	AEs
NCT04550260 (KUNLUN)	III	ESCC	Durvalumab + CRT (Cisplatin + 5-FU or Cisplatin + Capecitabine)	Placebo + CRT (Cisplatin + 5-FU or Cisplatin + Capecitabine)	PFS	OS, AEs
NCT04426955 (ESCORT-CRT)	III	ESCC	Camrelizumab + CRT (Paclitaxel + Cisplatin)	Placebo + CRT (Paclitaxel + Cisplatin)	PFS	OS, ORR, DoR, AEs
NCT04543617 (SKYSCRAPER-07)	III	ESCC	CRT (Platinum-based) + Tiragolumab + Atezolizumab (Cohort A) CRT (Platinum-based) + Tiragolumab Placebo+ Atezolizumab (Cohort B)	CRT (Platinum-based) + Tiragolumab Placebo + Atezolizumab Placebo	PFS, OS	ORR, DoR, QoL, AEs

CRT, chemoradiotherapy; ESCC, esophageal squamous cell carcinoma; EAC, esophageal adenocarcinoma; GEJC, gastroesophageal junction cancer; PFS, progression-free survival; ORR, objective response rate; DoR, duration of response; OS, overall survival; AEs, adverse events; EFS, event-free survival; QoL, quality of life.

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
