# Peer review of "Recent Advances in Combination of Immunotherapy and Chemoradiotherapy for Locally Advanced Esophageal Squamous Cell Carcinoma"

_cancers, 2022, doi:10.3390/cancers14205168_

Round 1

Reviewer 1 Report

Thank you for the article on the recent advances on chemoimmunotherapy in locally advanced esophageal squamous cell cancer. Well written article. I would advise to modify the tables to be more compact and easily readable. I would also suggest to include the metastatic setting which will be a more comprehensive review.  References need to be edited as some of them has more than 10 author names, reference 24 has a note before the author names which need to be removed. 

Reviewer 2 Report

The authors aimed to review the clinical application of immune checkpoint inhibitors (ICIs) in patients with locally advanced Esophageal squamous cell carcinoma (ESCC) and discuss potential biomarkers for predicting tumor response and survival outcomes.

The study covers some issues that have been overlooked in other similar topics. The structure of the manuscript appears adequate and well divided in the sections. Moreover, the study is easy to follow, but some issues should be improved. Some of the comments that would improve the overall quality of the study are:

a. Authors must pay attention to the technical terms acronyms they used in the text.

b. English language needs to be revised.

c. Conclusion Section: This paragraph is missing: please add it.

Reviewer 3 Report

Well written comprehensive overview
